# Criterion validity of the Saltin-Grimby Physical Activity Level Scale in adolescents. The Fit Futures Study

**Sigurd K. Beldo**[1]\*, **Nils Abel Aars**[2], **Tore Christoffersen**[3,4], **Anne-Sofie Furberg**[5,6], **Peder A. Halvorsen**[7], **Bjørge Herman Hansen**[8], **Alexander Horsch**[9], **Edvard H. Sagelv**[1], **Shaheen Syed**[9], **Bente Morseth**[1]

**1** School of Sport Sciences, Faculty of Health Sciences, UiT The Arctic University of Norway, Alta/Tromsø, Norway, **2** Nordlandssykehuset HF, Bodø, Norway, **3** Finnmark Hospital Trust, Alta, Norway, **4** Department of Health and Care Sciences, Faculty of Health Sciences, UiT The Arctic University of Norway, Alta/Tromsø, Norway, **5** Department of Microbiology and Infection Control, University Hospital of North Norway, Tromsø, Norway, **6** Faculty of Health Sciences and Social Care, Molde University College, Molde, Norway, **7** Department of Community Medicine, Faculty of Health Sciences, UiT The Arctic University of Norway, Tromsø, Norway, **8** Department of Sport Science and Physical Education, University of Agder, Kristiansand, Norway, **9** Department of Computer Science, UiT The Arctic University of Norway, Tromsø, Norway

\* sigurd.beldo@uit.no

**Data Availability Statement:** The data that support the findings of this study are available from The Fit Futures Study. However, confidentiality

## Abstract

### Background

The Saltin-Grimby Physical Activity Level Scale (SGPALS) is commonly used to measure physical activity (PA) in population studies, but its validity in adolescents is unknown. This study aimed to assess the criterion validity of the SGPALS against accelerometry in a large sample of adolescents. A secondary aim was to examine the validity across strata of sex, body mass index (BMI), parental educational level, study program and self-reported health.

### Methods

The study is based on data from 572 adolescents aged 15–17 years who participated in the Fit Futures Study 2010–11 in Northern Norway. The participants were invited to wear an accelerometer (GT3X) attached to their hip for seven consecutive days. We used Spearman's rho and linear regression models to assess the validity of the SGPALS against the following accelerometry estimates of PA; mean counts/minute (CPM), steps/day, and minutes/day of moderate-to-vigorous physical activity (MVPA).

### Results

The SGPALS correlated with mean CPM ($\rho = 0.40$, $p<0.01$), steps/day ($\rho = 0.35$, $p<0.01$) and MVPA min/day ($\rho = 0.35$, $p<0.01$). We observed no differences between correlations within demographic strata (all $p>0.001$). Higher scores on SGPALS were associated with a higher CPM, higher number of steps per day and more minutes of MVPA per day, with the following mean differences in PA measurements between the SGPALS ranks: CPM increased by 53 counts (95% CI: 44 to 62), steps/day increased by 925 steps (95% CI: 731

requirements according to Norwegian law prevents sharing of individual patient level data in public repositories. The legal restriction on data availability are set by the Fit Futures Data and Publication Committee in order to control for data sharing, including publication of datasets with the potential of reverse identification of de-identified sensitive participant information. Data can be made available from the The Fit Futures Study upon application. To apply for data, please contact the Fit Futures at fitfutures@uit.no.

**Funding:** The work of Edvard H Sagelv was funded by the Population Studies in the High North (Befolkningsundersøkelser i nord: BiN, internally funded, no grant number). https://uit.no/research/bin The funders had no role in study design, data collection and analysis, decision to publish, or preparation of the manuscript.

**Competing interests:** The authors have declared that no competing interests exist.

**Abbreviations:** BMI, Body Mass Index; CPM, Count per minute; FF1, Fit Futures 1; MVPA, Moderate to vigorous Activity; PA, Physical activity; PAQ, Physical activity questionnaire; QCAT, Quality Control & Analysis Tool; SGPALS, Saltin-Grimby Physical Activity Level Scale; VM, Vector Magnitude; WHO, World Health Organization.

to 1118), and MVPA by 8.4 min/day (95% CI: 6.7 to 10.0). Mean difference between the highest and lowest SGPALS category was 2947 steps/day (6509 vs. 9456 steps/day) and 26.4 min/day MVPA (35.2 minutes vs 61.6 minutes).

## Conclusion

We found satisfactory ranking validity of SGPALS measured against accelerometry in adolescents, which was fairly stable across strata of sex, BMI, and education. However, the validity of SGPALS in providing information on absolute physical activity levels seem limited.

## Introduction

Low levels of physical activity (PA) in adolescence are associated with an increased risk of obesity and non-communicable diseases in adulthood [1, 2]. PA levels in childhood and adolescence seem to decline with increasing age [3] and tend to track into adulthood [4]. Consequently, surveillance of PA in childhood and adolescence is vital to inform public health policies aimed at increasing or maintaining PA levels in childhood and adolescence [4].

In population-based studies, questionnaires are the most common measure of PA, being inexpensive, practical and a quick and scalable method for collecting data. However, self-reported PA is likely influenced by recall and social desirability bias, which may introduce misclassification and influence the validity of self-reported PA [5–8]. Validation of self-reported PA instruments is therefore crucial for interpreting prevalence estimates of PA and associations between PA and health outcomes [9].

One of the most frequently used physical activity questionnaires (PAQs) in Scandinavia is the Saltin-Grimby Physical Activity Level Scale (SGPALS), introduced by Saltin and Grimby in 1968 [10, 11]. The SGPALS includes four hierarchical ranks of PA [10] and is included in numerous population studies [11–14]. The SGPALS is predominantly used in adult cohorts but is also included in some adolescent cohort studies [15, 16]. In adults, higher SGPALS ranks has been shown to represent higher criterion measure estimates, such as accelerometry and cardiorespiratory fitness [11]. In children and adolescents, differences between self-reported and device measured PA has been reported [17, 18]. A modified Motric Module PAQ underestimated LPA and MVPA during school hours, but overestimated leisure-time activity, compared to accelerometry [17]. To our knowledge, the SGPALS has not been compared to such criterion measures in adolescents.

In adults, previous research indicates that the agreement between self-reported and device measured PA may differ within strata, showing a higher discordance among individuals with low education [19–21]. Moreover, men are found to report higher PA than women despite accumulating similar device measured PA [20] but not consistently [19, 22, 23].

In contrast to adults, higher education groups show higher differences between self-reported and device measured PA than lower education groups in adolescent samples [24]. Although not consistent [25], sex differences between self-reported and device measured PA also seem evident in adolescents [18, 26, 27]. Moreover, there are inconsistent findings in validity of self-reported PA by BMI groups in adolescents, where some studies found no differences by BMI groups [18], while others report BMI to influence the discrepancies between self-reported and device measured PA [28]. As several factors may influence the validity of self-reported PA in adolescents, further exploration on demographic factors that can influence

discrepancies between self-reported, especially SGPALS, and device measured PA is warranted.

Thus, the aim of this study was to explore validity of the SGPALS in a sample of Norwegian adolescents. A secondary aim was to examine the validity by strata of sex, BMI, parental educational level, self-reported health status and school program.

## Materials and methods

### Design and participants

The Fit Futures Study (FF) is a population-based cohort study of adolescents in Northern Norway [15, 16]. We used data from the first survey of the FF (FF1), collected between September 2010 and April 2011. All first-year high school students (n = 1117) from one urban (Tromsø) and one rural (Balsfjord) municipality in Northern Norway were invited to participate, of which 1038 (92.7%) attended. The participants attended a half-day visit at the Clinical Research Department at the University Hospital of Northern Norway, Tromsø, and all procedures were performed by trained research technicians. The data collection included electronic questionnaires, clinical examinations and accelerometer measurements. The accelerometers (ActiGraph GT3X, ActiGraph, Pensacola, FL, United States) were handed out to the participants with instructions to wear the device on their right hip for seven consecutive days. In the present study we excluded participants with accelerometer wear time <10 hours for at least 4 days (n = 427) and those aged $\geq$ 18 years (n = 38). The final sample included 572 participants with valid accelerometer wear time and complete data on the SGPALS questionnaire. A larger proportion of girls than boys (68% vs 52%, p < 0.001) and those studying general studies (70% vs 58% of both vocational and sports, p < 0.001) provided valid accelerometry data while distribution of parental level of education (p = 0.33), BMI (p = 0.41) and self-reported health (p = 0.81) did not differ between those with and without valid accelerometry data (S1 Table).

Participants aged 16 years or above signed a written informed consent. Participants under 16 years signed with written permission from their legal guardians. The Regional Committee for Medical and Health Ethics approved the study (2012/1663/REK Nord).

### Socio-demographic variables

Weight and height were measured on a Jenix DS-102 stadiometer (Dong Sahn Jenix co Ltd, Seoul, Korea), an automatic electronic scale. Weight was measured in kilograms (kg) with a precision of 0.1 kg and height in meters (m) to the nearest 0.1 cm. BMI was calculated as kg divided by the square height (kg/m$^2$). According to the International Obesity Task Force (iso-BMI), at the age of 16 the cut-off for overweight is 23.9 kg/m$^2$ for boys and 24.37 kg/m$^2$ for girls [29]. As iso-BMI and adult cut-offs for BMI become more similar by increasing age, BMI was calculated according to adults' cut-offs and categorized as normal weight ($<$ 25 kg/m$^2$), and overweight and obese ($\geq$ 25 kg/m$^2$). Socioeconomic status was determined by questionnaire data on the parent with the highest level of education, categorized as either; 1) Do not know, 2) Primary/high school, 3) University $<$ 4 years, and 4) University 4 $\geq$ years. The participants rated their self-perceived health according to the question: *«How do you in general consider your own health to be*?*»*, with five alternatives: 1) Very poor, 2) Poor, 3) Neither good nor poor, 4) Good, or 5) Excellent. Only four participants rated their health as very poor, thus we categorized 1) Very poor and 2) Poor into "1) Very poor/poor". Information on study program (vocational, general studies or sports) [30] was retrieved from the schools' student database.

In Norway, first year of upper secondary school means the 11[th] year of Norwegian school attendance, where the students can choose between different study programs. About 38% choose general studies, 6% choose sports specialization, and the remaining students choose

between 11 different vocational studies such as health programs, technical programs, maritime programs, creative schools and economic and administrative programs [31].

## The Saltin-Grimby Physical Activity Level Scale—SGPALS

Participants answered the SGPALS by stating their PA level according to four hierarchical levels [10, 11]. Compared with the original wording by Saltin and Grimby in 1968 designed for adults [10], the participants in this study answered a slightly modified version with examples of activities suited for adolescents (Table 1), and with a duration requirement also for level 3 (in addition to level 2). This has later been the version recommended by Grimby and colleagues [11, 32].

## Accelerometer data collection and processing

The ActiGraph GT3X records accelerations in three axes (axial, coronal and sagittal). The devices were initialized using the manufacturer's software (ActiLife, LLC, Pensacola, FL, USA) with 30 Hz sampling frequency, and set to record data from when the ActiGraph was attached to the hip and until 23:59 on day 8. The ActiLife software was used to download the accelerometer data using the normal (default) filter to aggregate raw acceleration data into 10-seconds epochs using a proprietary algorithm designed by the manufacturer. The data were further analyzed using the Quality Control & Analysis Tool (QCAT), a custom-made software developed in Matlab (The MathWorks, Inc, Natick, MA, USA). The 10-second epochs were summed to 60 seconds, and the first day of measurements was excluded from further analyses to reduce reactivity [33].

Wear time was calculated from triaxial vector magnitude (the square root of the sum of squared activity) counts per minute (CPM) as described by Hecht et al. [34], based on the following three criteria; 1) A vector magnitude value (VMU) in counts per minute (CPM) > 5; 2) Of the following 20 minutes, at least 2 minutes have VMU CPM values > 5; and 3) Of the preceding 20 minutes, at least 2 minutes have VMU CPM values > 5. If at least 2 of the criteria were positive, the 1-minute epoch was considered as wear-time. All other minutes were defined as non-wear time.

We expressed volume estimates of PA as mean uniaxial CPM per day, number of steps per day and moderate-to-vigorous physical activity (MVPA). The step count was derived from the vertical axis using a proprietary algorithm from the manufacturer. MVPA was defined as a $CPM \geq 1952$ [35], measured in minutes per day (min/day).

**Table 1. Saltin-Grimby Physical Activity Level Scale (SGPALS).**

|  | Leisure Time Physical Activity Level |
|---|---|
| Question | Which description fits best regarding your physical activity level in leisure time during the last year? |
| Answering alternative 1 | Almost completely inactive: |
|  | *"Sitting by the PC/TV, reading or other sedentary activity"* |
| Answering alternative 2 | Moderately active: |
|  | *"Walking, cycling, or other forms of exercise at least 4 hours per week (here, you should also consider transport to/from school, shopping, Sunday strolls etc.)"* |
| Answering alternative 3 | Highly active: |
|  | *"Participation in recreational sports, heavy outside activity, shoveling snow etc. at least 4 hours per week"* |
| Answering alternative 4 | Vigorously active: |
|  | *"Participation in hard training or sports competitions regularly several times a week".* |

## Statistical analyses

Participants who did not meet our wear time criterion of at least four days with $\geq$ 10 hours of activity [36] were excluded from the analysis. All accelerometer estimates (CPM, steps, and MVPA) were considered normally distributed by visual inspection of histograms and QQ-plots. We used independent t-tests to assess differences in accelerometry wear time between boys and girls, and between under- and normal weight and overweight and obese participants. Differences in accelerometer wear time between study programs, parental education and self-reported health status were assessed by univariate analyses of variance (ANOVA). We also used ANOVAs to assess the association between indices of device-measured PA (CPM, steps, and MVPA) and the SGPALS. We used Spearman's rho ($\rho$) to assess the ranked correlation between the SGPALS and accelerometer estimates of PA (mean CPM, mean steps/day and min/day MVPA) in total and in strata of sex, BMI, parental level of education, self-reported health, and study program. We visually inspected scatter plots following our correlation analyses to identify outliers. We used Fisher´s $\rho$ to z transformation to compare rho correlations within demographic strata, as previously done by others [37]. To decrease false discovery rates, we adjusted the p-values from Spearman´s rho, and for comparison between rho´s, according to the Benjamin-Hochberg method [38] with 25% false discovery rate. A coefficient ($\rho$) of 0.00 to 0.10, 0.10 to 0.39, 0.40 to 0.69 and $\geq$ 0.70 was considered a negligible, weak, moderate and strong correlation, respectively [39]. Alpha was set to $p < 0.05$. All data are presented as mean ± standard deviation (SD), mean with 95% confidence interval (CI) or as frequency (percentage). All analyses were performed using the Statistical Package for Social Science (SPSS Version 25, International Business Machines Corporation, USA).

## Results

The descriptive characteristics of participants are presented in Table 2. Among the 253 boys and 319 girls in this study, mean BMI was 22.4 kg/m$^2$ (both sexes) and the mean age was 16.0 and 16.1 years, respectively. Among the 572 participants, 98 (17.1%) classified themselves in the first category of the SGPALS, 197 (34.4%) in the second category, 164 (28.7%) in the third and 113 (19.8%) in the last category. Girls were more likely to report lower self-reported health status than boys (p = 0.26). There were differences in wear time per valid day between sexes (p = 0.01), but not between BMI categories (p = 0.83), study program (p = 0.35), parental education (p = 0.23) and self-reported health status (p = 0.38) (S2 Table). Mean MVPA was 44.8 (SD 21.7) minutes per day, mean CPM 340.8 (SD 123.0) and mean number of steps per day was 7875 (SD 2508).

The distribution of CPM, steps and MVPA is illustrated by box plots in Fig 1. We observed statistically significant increases in all indices of accelerometer measured PA by increasing SGPALS levels (all p < 0.001). Mean difference between the lowest and highest SGPALS categories was 163 CPM (278 vs. 441 mean CPM), 2947 steps/day (6509 vs. 9456 steps/day) and 27 min/day MVPA (35 minutes vs 62 minutes) (Table 3).

The SGPALS was positively correlated with steps/day ($\rho$ = 0.35, p<0.01), min/day MVPA ($\rho$ = 0.35, p<0.01), and mean CPM ($\rho$ = 0.40, p<0.01) (Table 4). We observed no differences in correlations between socio-demographic strata (all p>0.001).

## Discussion

In this population-based validation study among Norwegian adolescents, we found positive associations between self-reported PA measured by the SGPALS and accelerometer-measured PA. Although correlations between the SGPALS and accelerometer measured PA in general were weak, the SGPALS was able to correctly rank accelerometer-measured PA, as we

**Table 2. Characteristics of boys and girls, the Fit Futures Study 2010–2011.**

| | All | Girls | Boys | SGPALS | | | |
|---|---|---|---|---|---|---|---|
| | (n = 572) | (n = 319) | (n = 253) | | | | |
| | | | | 1 | 2 | 3 | 4 |
| Age (years) | 16.1±0.4 | 16.1±0.4 | 16.0±0.4 | 16.1±0.4 | 16.1±0.4 | 16.1±0.4 | 16.1±0.3 |
| Height (cm) | 170.4±8.8 | 165.2±6.4 | 177.0±6.8 | 170.9±8.8 | 169.2±8.5 | 170.1±8.6 | 172.6±9.3 |
| Weight (kg) | 65.3±13.6 | 61.2±11.3 | 70.4±14.4 | 65.6±15.1 | 65.5±13.5 | 65.0±14.6 | 65.0±10.7 |
| BMI (kg/m$^2$) | 22.4±4.0 | 22.4±3.8 | 22.4±4.2 | 22.4±4.5 | 22.8±4.1 | 22.4±4.4 | 21.7±2.4 |
| **BMI category n (%)** | **570 (100)** | **317 (100)** | **253 (100)** | **1** | **2** | **3** | **4** |
| Underweight or normal weight * | 457 (80.2) | 260 (82.0) | 197 (77.9) | 71 (15.5) | 148 (32.4) | 139 (30.4) | 99 (21.7) |
| Overweight or obese | 113 (19.8) | 57 (18.0) | 56 (22.1) | 26 (23.0) | 48 (42.5) | 25 (22.1) | 14 (12.4) |
| **Study specialization n (%)** | **572 (100)** | **319 (100)** | **253 (100)** | **1** | **2** | **3** | **4** |
| Vocational | 238 (41.7) | 108 (33.8) | 130 (51.4) | 57 (23.9) | 107 (45.0) | 53 (22.3) | 21 (8.8) |
| General | 273 (47.6) | 184 (57.7) | 89 (35.2) | 41 (15.0) | 87 (31.9) | 99 (36.3) | 46 (16.8) |
| Sports | 61 (10.7) | 27 (8.5) | 34 (13.4) | 0 | 3 (4.9) | 12 (19.7) | 46 (75.4) |
| **Parents' education n (%)** | **570 (100)** | **318 (100)** | **252 (100)** | **1** | **2** | **3** | **4** |
| Do not know | 113 (19.8) | 52 (16.4) | 61 (24.2) | 24 (21.2) | 39 (34.5) | 30 (26.5) | 20 (17.7) |
| Primary/high school | 167 (29.3) | 89 (28.0) | 78 (31.0) | 34 (20.4) | 65 (38.9) | 46 (27.5) | 22 (13.2) |
| University <4 years | 115 (20.2) | 72 (22.6) | 43 (17.0) | 11 (9.6) | 41 (35.7) | 29 (25.2) | 34 (29.6) |
| University ≥4 years | 175 (30.7) | 105 (33.0) | 70 (27.8) | 28 (16.0) | 51 (29.1) | 59 (33.7) | 37 (21.1) |
| **Self-perceived health n (%)** | **569 (100)** | **317 (100)** | **252 (100)** | **1** | **2** | **3** | **4** |
| Very poor/poor | 30 (5.3) | 18 (5.7) | 12 (4.8) | 12 (40.0) | 11 (36.7) | 5 (16.7) | 2 (6.7) |
| Neither good nor poor | 119 (20.9) | 61 (19.2) | 58 (23.0) | 36 (30.3) | 55 (46.2) | 18 (15.1) | 10 (8.4) |
| Good | 276 (48.5) | 170 (53.6) | 106 (42.1) | 40 (14.5) | 98 (35.5) | 91 (33.0) | 47 (17.0) |
| Excellent | 144 (25.3) | 68 (21.5) | 76 (30.1) | 10 (6.9) | 31 (21.5) | 50 (34.7) | 53 (36.8) |

Data are mean ± standard deviation. SGPALS = Saltin-Grimby Physical Activity Level Scale (Which description fits best regarding your physical activity level in leisure time during the last year? 1 Almost completely inactive "Sitting by the PC/TV, reading, or other sedentary activity". 2 Moderately active "Walking, cycling or other forms of exercise at least 4 hours per week (here you should also consider transport to/from school, shopping, Sunday strolls etc.". 3 Highly active "Participation in recreational sports, heavy outside activity, shoveling snow etc. at least 4 hours per week". 4 Vigorously active "Participation in hard training or sports competitions regularly several times a week". BMI = Body mass index.

*Cut-off value<25.

observed a notable and gradual increase in accelerometry measures for each increase in SGPALS levels.

The SGPALS correlated moderately with accelerometer-measured mean CPM, steps/day and min/day of MVPA. These observations are consistent with previous studies in adults [40–43], where the ranking ability of the SGPALS has been demonstrated against accelerometry [41–43] and cardiorespiratory fitness measures in adults [40–44]. In our study of adolescents, the SGPALS demonstrated similar ranking ability of PA levels. For example, for every increase in SGPALS level, steps per day increased with ~1000 steps and MVPA with ~ 8 minutes per day. This sums up to ~7000 steps and ~60 minutes of MVPA extra per week if individuals increase their PA by one SGPALS level. Such increases would have relevant impact on public health and thus highlights the SGPALS´ ranking ability at the population level. Similar increases in step count by higher SGPALS ranks are found in adults, while increases in MVPA seem to be lower (~2 min by increasing SGPALS rank) [42].

In general, the correlations in our study and that of others [40–43] are modest, which highlight the imprecision associated with self-reported PA [45] and shows that the SGPALS is unable to precisely reflect accelerometry estimates of PA. Nevertheless, 95% of those reporting

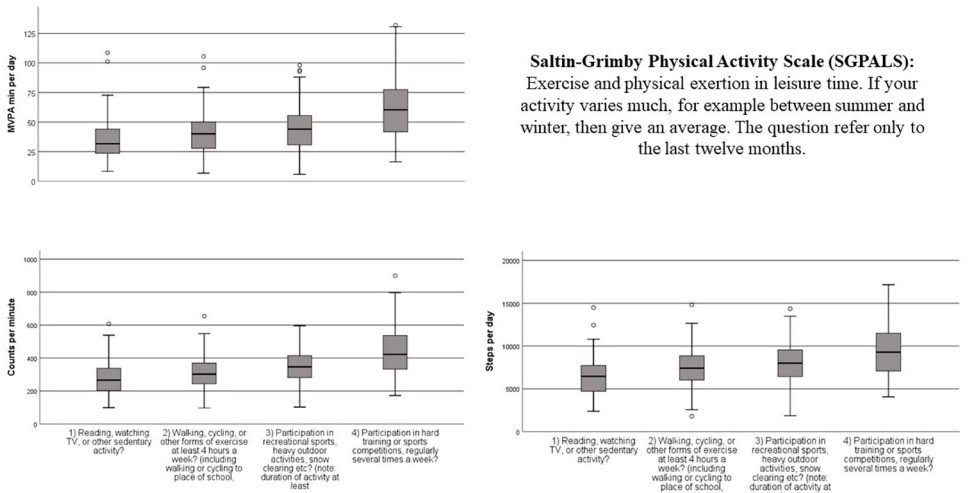

**Fig 1. Box plot with median, interquartile range, maximum and minimum with outliers of CPM, steps and MVPA per day by SGPALS ranks.** The Fit Futures Study 2010–2011.

to be inactive (rank 1) in the SGPALS were also physically inactive in accelerometry estimates (<60 minutes of MVPA), indicating that the SGPALS is fairly good at classifying inactive individuals (Table 3). Although the proportion of individuals classified as active by the accelerometer increases by increasing rank, it seems that in the higher ranks, the precision in classifying active vs. inactive individuals decreases (Table 3). Although the accuracy of PA volume and intensity is limited when using the SGPALS, crude ranking of self-reported PA at population level is valuable [45]. The SGPALS is shorter than most recall-PAQs, which could make it more appealing to researchers, especially when planning population studies.

The SGPALS is sometimes labelled a "global questionnaire", aiming to provide a prompt overview of the level of PA. Another common type of questionnaire is the "short recall PAQ", providing a quick assessment of the total volume of PA, often classified by intensity level (often moderate and vigorous PA) or by domains (work related PA, leisure time PA, or transportation). Examples are "The School Health Action, Planning and Evaluation System

**Table 3. Accelerometer measured physical activity according to the Saltin-Grimby Physical Activity Level Scale (SGPALS).** The Fit Futures Study 2010–2011.

| | SGPALS (n = 572) | | | |
| --- | --- | --- | --- | --- |
| | *Inactive* | *Moderately active* | *Highly active* | *Vigorously active* |
| | *(n = 98, 17.1%)* | *(n = 197, 34.4%)* | *(n = 164, 28.7%)* | *(n = 113, 19.8%)* |
| Mean CPM per day* (95% CI) | 277.9 | 307.1 | 348.6 | 441.4 |
| | (256.1–299.7) | (291.7–322.4) | (331.8–365.5) | (421.1–461.7) |
| Steps per day* (95% CI) | 6509 | 7481 | 8060 | 9456 |
| | (6046–6971) | (7153–7808) | (7703–8418) | (9023–9889) |
| MVPA (min/day)* (95% CI) | 35.2 | 39.8 | 44.7 | 61.6 |
| | (31.3–39.1) | (37.1–42.6) | (41.7–47.8) | (57.9–65.2) |
| N (%) | 98 (100) | 197 (100) | 164 (100) | 113 (100) |
| Meeting PA guidelines | 5 (5.1) | 22 (11.2) | 34 (20.7) | 57 (50.4) |

*Statistically significant difference between categories (between-subject difference): p<0.001. Data are unadjusted mean and 95%CI. CPM = counts per minute, Steps = steps per day, MVPA = moderate-to-vigorous physical activity, CI = confidence interval.

**Table 4. Spearman rank correlations between SGPALS ranks and accelerometer-measured physical activity.** The Fit Futures Study 2010–2011.

|  | Mean CPM | Steps | MVPA |
|---|---|---|---|
| All (n = 572) | **0.40*** | **0.35*** | **0.35*** |
| **Sex** |  |  |  |
| Boys (n = 253) | **0.40*** | **0.37*** | **0.34*** |
| Girls (n = 319) | **0.41*** | **0.33*** | **0.38*** |
| **BMI category** |  |  |  |
| Underweight or normal weight (n = 457) | **0.43*** | **0.35*** | **0.38*** |
| Overweight or obese (n = 113) | **0.27*** | **0.32*** | **0.20** |
| **Study specialization** |  |  |  |
| Vocational (n = 238) | **0.30*** | **0.31*** | **0.26*** |
| General (n = 273) | **0.33*** | **0.23*** | **0.23*** |
| Sports (n = 61) | 0.25 | 0.20 | 0.23 |
| **Parents' education** |  |  |  |
| Do not know (n = 113) | **0.37*** | **0.40*** | **0.35*** |
| Primary/high school (n = 167) | **0.26*** | **0.25*** | **0.24*** |
| University <4 years (n = 115) | **0.47*** | **0.38*** | **0.41*** |
| University ≥4 years (n = 175) | **0.48*** | **0.38*** | **0.41*** |
| **Self-perceived health** |  |  |  |
| Very poor/poor (n = 30) | **0.40** | **0.39** | 0.35 |
| Neither good nor poor (n = 119) | **0.30*** | **0.28*** | **0.24*** |
| Good (n = 276) | **0.31*** | **0.25*** | **0.28*** |
| Excellent (n = 144) | **0.47*** | **0.42*** | **0.43*** |

SGPALS = Saltin-Grimby Physical Activity Level Scale. BMI = body mass index, CPM = counts per minute,
Steps = steps per day, MVPA = moderate-to-vigorous physical activity, bold numbers indicate significant Spearman's
rho at p<0.05

*Significant Spearman's rho at p<0.01.

(SHAPES) [28, 46], International Physical Activity Questionnaire (IPAQ) [22, 47, 48] and WHO Health Behavior in School-aged Children (HBSC) [49]. These recall questionnaires yield more information than global questionnaires, however, this also introduces a risk of lower precision. For example, more questions and exceeding details may hamper participants' ability to recall all details associated with participation in physical activity. Moreover, there may be difficulties related to the comprehension of the concepts of "moderate" and "vigorous" PA and in recalling normal activities such as walking or sitting, and calculating total duration [50]. The SGPALS showed correlations with accelerometer measures that are comparable with other PAQs validated in adolescents [51, 52]. Consequently, although the accuracy of PA volume and intensity is limited when using the SGPALS, crude ranking of self-reported PA at population level is valuable [45], and presents SGPALS as a viable option when choosing PAQs as it is relatively easy to answer and obtain fairly accurate PA estimates.

The ranking ability of the SGPALS was similar across various socio-demographic strata. This contrasts with some previous studies comparing other PA questionnaires in adolescents against accelerometry measured PA by sex [18, 26, 27], parental education [24], and categories of BMI [18], although some have reported no differences in ranking ability between sex and BMI groups [17, 18].

Inconsistent findings may be explained by differences in the distribution of socio-demographic variables or by measurement properties in the PA questionnaires included. Most PA

questionnaires ask participants to report minutes in different intensities [17–21, 24–28], while the SGPALS include four crude groups representing PA in the last year. Considering inconsistent findings between socio-demographic strata in previous studies [17–21, 24–28], multiple item PAQs may inherently influence measurement precision due to adolescents´ recall abilities. Our findings of stable correlations across strata suggest the SGPALS to be fairly robust in ranking PA levels without much influence from socio-demographical characteristics in adolescents.

### Strengths and limitations

To our knowledge, this is the first study to assess the validity of the leisure time SGPALS in adolescents, as few other studies have used accelerometry to measure PA in larger samples in this particular age group. Moreover, Fit Futures had a high participation proportion (93%), although missing accelerometer data resulted in a considerably large proportion that did not provide valid accelerometer wear time; thus, our results may be influenced by selection bias. Consequently, one should be cautious when interpreting the results. However, in a recent publication based on the same population from Fit Futures, missing accelerometer data were imputed and a sensitivity analysis showed that the participants with missing accelerometer data did not differ significantly from the participants with valid data [53].

Further, the accelerometer assessment over seven days was not time-aligned with the SGPALS [10, 11, 49]; the SGPALS addresses habitual PA (over the last year) and participants completed the instrument at the start of the accelerometer wear period. However, PA instruments are in general designed to capture the habitual PA level [54], with the SGPALS [55, 56] showing acceptable reliability (moderate Kappa ~0.5–0.6), as does four days of $\geq 10$ hours of accelerometer assessment (intraclass correlation: 0.8) [49, 54]. As the SGPALS was filled out immediately before wearing the accelerometer, this may have introduced reactivity [33]. In an attempt to overcome the potential for reactivity, we excluded the first day of accelerometry recording.

Furthermore, this study validated the leisure time PA part of the SGPALS, including modes of transportation to/from school, while the accelerometer assessment is not limited to leisure time. The occupational time SGPALS [10] was not included in this study of adolescents as it is not relevant for this age group attending high school.

## Conclusion

Our study adds to building evidence for satisfactory ranking validity of SGPALS measured against accelerometry in adolescents, and the validity is fairly stable across strata of sex, BMI, and education. However, the validity of SGPALS in providing information on absolute physical activity levels is limited.

### Implications for public health and future research

In a public health perspective, increasing PA is more important among those who are inactive, as changing PA levels from low levels seem to yield more health effects than increasing from higher levels of PA [57]. The current study provides evidence to support the use of SGPALS as a low cost and time efficient tool to identify the least active adolescents. Future research may involve exploring the stability of the SGPALS stability, compared to other self-report measurements. Another potential research area could be comparing the validity of SGPALS among youth to that among adults to elicit if age does influence the validity of the instrument.

## Supporting information

**S1 Table. Distribution of valid and invalid accelerometry wear time.**
(DOCX)

**S2 Table. Accelerometry wear time by BMI, study specialization, parental education and self-perceived health.** The Fit Futures Study 2010–2011.
(DOCX)

## Acknowledgments

The authors are grateful for the contribution by the participants in the Fit Futures study. We thank the research technicians at the Clinical Research Department, University Hospital of North Norway for facilitating data collection in the Fit Futures study.

## Author Contributions

**Conceptualization:** Sigurd K. Beldo, Nils Abel Aars, Edvard H. Sagelv, Bente Morseth.

**Data curation:** Sigurd K. Beldo, Nils Abel Aars, Tore Christoffersen, Edvard H. Sagelv.

**Formal analysis:** Sigurd K. Beldo, Nils Abel Aars, Tore Christoffersen, Peder A. Halvorsen, Edvard H. Sagelv, Bente Morseth.

**Funding acquisition:** Anne-Sofie Furberg.

**Investigation:** Sigurd K. Beldo, Anne-Sofie Furberg, Alexander Horsch, Edvard H. Sagelv, Shaheen Syed, Bente Morseth.

**Methodology:** Sigurd K. Beldo, Nils Abel Aars, Tore Christoffersen, Peder A. Halvorsen, Bjørge Herman Hansen, Edvard H. Sagelv, Bente Morseth.

**Project administration:** Anne-Sofie Furberg, Bente Morseth.

**Software:** Alexander Horsch, Shaheen Syed.

**Supervision:** Peder A. Halvorsen, Bjørge Herman Hansen, Alexander Horsch, Bente Morseth.

**Validation:** Anne-Sofie Furberg, Bjørge Herman Hansen, Alexander Horsch, Shaheen Syed, Bente Morseth.

**Visualization:** Sigurd K. Beldo.

**Writing – original draft:** Sigurd K. Beldo, Edvard H. Sagelv.

**Writing – review & editing:** Sigurd K. Beldo, Nils Abel Aars, Tore Christoffersen, Anne-Sofie Furberg, Peder A. Halvorsen, Bjørge Herman Hansen, Alexander Horsch, Edvard H. Sagelv, Shaheen Syed, Bente Morseth.

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
