## [Decision Letter · Decision Letter 0]

19 May 2021

PONE-D-21-01038

Criterion validity of the Saltin-Grimby Physical Activity Level Scale in adolescents. The Fit Futures Study.

PLOS ONE

Dear Dr. Beldo,

Thank you for submitting your manuscript to PLOS ONE. After careful consideration, we feel that it has merit but does not fully meet PLOS ONE’s publication criteria as it currently stands. Therefore, we invite you to submit a revised version of the manuscript that addresses the points raised during the review process.

The reviewers raised a number of concerns, including the presentation/flow of the manuscript and the thoroughness of the Discussion, as well the statistical analyses and the presentation of data. Their comments can be viewed in full, below.

We look forward to receiving your revised manuscript.

Kind regards,

Natasha McDonald, PhD

Associate Editor

PLOS ONE

Journal Requirements:

2) Please ensure that you have obtained any necessary permissions (if applicable) to use and reproduce the SGPALS scale in the current manuscript.

3)  We note that you have indicated that data from this study are available upon request. PLOS only allows data to be available upon request if there are legal or ethical restrictions on sharing data publicly. For information on unacceptable data access restrictions, please see http://journals.plos.org/plosone/s/data-availability#loc-unacceptable-data-access-restrictions.

4) Your ethics statement should only appear in the Methods section of your manuscript. If your ethics statement is written in any section besides the Methods, please delete it from any other section.

Reviewers' comments:

Reviewer's Responses to Questions

**Comments to the Author**

1. Is the manuscript technically sound, and do the data support the conclusions?

Reviewer #1: Partly

Reviewer #2: Partly

2. Has the statistical analysis been performed appropriately and rigorously? 

Reviewer #1: No

Reviewer #2: Yes

3. Have the authors made all data underlying the findings in their manuscript fully available?

Reviewer #1: Yes

Reviewer #2: No

4. Is the manuscript presented in an intelligible fashion and written in standard English?

Reviewer #1: No

Reviewer #2: Yes

5. Review Comments to the Author

Reviewer #1: PONE-D-21-01038_Rview Comments

This study aimed to assess the criterion validity of the Saltin-Grimby Physical Activity Level Scale (SGPALS) against accelerometry in a large sample of adolescents. In addition, the validity in strata of sex, body mass index (BMI), parental educational level, study program and self-reported health has also been examined.

1. I highly suggest authors to consider further revise the introduction section by including more detailed background information, rationales for why validating the SGPALS, the difference between the SGPALS, and the Godin Leisure-Time Exercise Questionnaire, and the International Physical Activity Questionnaires (IPAQ). The length of the current introduction is brief and lacks supporting and transition sentences. For example, there is a little bit logic gap from the first paragraph to the second paragraph of the introduction section. Even within the first paragraph, there are only three sentences the three sentences did not link to each other in a correct way. There is also a lack of a section focused on the aims of the study with hypotheses and so on. Given there is a second aim of the study, background information on justifying the necessary to conduct difference analyses on these variables are needed. Overall, one page for the introduction is not sufficient, suggest increasing the length to three pages.

2. I also have a concern regarding the standard for classifying participants as minors and adults using 16 as the cutoff criteria. The question is that why not using 18 as the cutoff point but choosing 18.

3. It seems that this is a secondary data analysis of an existing project called “The Fit Futures Study (FF)”. I think this needs to be specific across the whole study.

4. Page 5, the subheading is titled as “Covariates”. I am not sure whether this is correct as they were not controlled in regression models or SEM but more of a key social demographic variable that that been used to compare the difference for the SGPALS scores.

5. For the time frame of the first question in the scale “Which description fits best regarding your physical activity level in leisure time during the last year?” The whole year might be exceptionally long that leads to memory bias and inaccuracy in recall. I doubt this time frame.

6. For discussion, four brief paragraphs are not enough. Findings of the current study should be further explained with support from previous studies. In addition, findings of the current study should be compared with previous studies for consistency and inconsistency.

7. When talking about the “Strengths and limitations” of the current study, it is better that future studies should be mentioned. For example, the last point at Lines 268-270.

8. The last section of the study is not written in a well-summarized and clear way. Needs to be further revised to make it clear.

Reviewer #2: I have attached my full review in pdf document and now I am writing until I get to the minimum number of characters for this box on the website. The limit is 200 characters. Thank you for submitting and allowing me to read your work. I hope my comments help.

6. PLOS authors have the option to publish the peer review history of their article (what does this mean?). If published, this will include your full peer review and any attached files.

Reviewer #1: No

Reviewer #2: No

---

## [Author Response · Author response to Decision Letter 0]

18 Nov 2021

Response to reviewers

All line-references are to the line number in the revised tracked changes document.

Reviewer 1:

Reviewer comment 1: I highly suggest authors to consider further revise the introduction section by including more detailed background information, rationales for why validating the SGPALS, the difference between the SGPALS, and the Godin Leisure-Time Exercise Questionnaire, and the International Physical Activity Questionnaires (IPAQ). The length of the current introduction is brief and lacks supporting and transition sentences. For example, there is a little bit logic gap from the first paragraph to the second paragraph of the introduction section. Even within the first paragraph, there are only three sentences the three sentences did not link to each other in a correct way. There is also a lack of a section focused on the aims of the study with hypotheses and so on. Given there is a second aim of the study, background information on justifying the necessary to conduct difference analyses on these variables are needed. Overall, one page for the introduction is not sufficient; suggest increasing the length to three pages.

Response: Thank you. 

Comparing PA questionnaires: Although the Godin Leisure-time Exercise Q and the IPAQ are commonly used questionnaires in many studies, these questionnaires are not included in the Fit-Futures study. Thus, providing background information on these questionnaires seems beyond the scope of this study, as it does not directly relate to our study or our aims. 

Coherence and transition: We have made an effort to revise the text, with proper transitions between paragraphs. 

Rationale for aims and length of introduction: We have now changed our introduction to more explicitly highlight the reason for validating the SGPALS in adolescents, line 77-90. Additionally, we have provided a new paragraph justifying the choice of stratification variables in our study. Please see line 92-106. This has added substantially to the length of the introduction.

Reviewer comment 2: I also have a concern regarding the standard for classifying participants as minors and adults using 16 as the cutoff criteria. The question is that why not using 18 as the cutoff point but choosing 18.

Response: According to the Norwegian Patient and User Rights Act, a person is of a legal age in relation to health service rights when they turn 16 years (Pasient- og brukerrettighetsloven, 

LOV-1999-07-02-63, Retrieved from Lovdata.no; https://lovdata.no/dokument/NL/lov/1999-07-02-63. 

Reviewer comment 3: It seems that this is a secondary data analysis of an existing project called “The Fit Futures Study (FF)”. I think this needs to be specific across the whole study.

Response: The Fit Futures Study (FF) is an ongoing longitudinal population-based cohort study, with survey waves conducted in 2010-11, 2012-13 and a third is ongoing (2021). The analyses in this manuscript are performed on the first survey performed in 2010-11 (FF1). Please see line 118-120.

Reviewer comment 4: Page 5, the subheading is titled as “Covariates”. I am not sure whether this is correct as they were not controlled in regression models or SEM but more of a key social demographic variable that that been used to compare the difference for the SGPALS scores.

Response: Thank you. We have accordingly changed the subheading to socio-demographic variables.

Reviewer comment 5: For the time frame of the first question in the scale “Which description fits best regarding your physical activity level in leisure time during the last year?” The whole year might be exceptionally long that leads to memory bias and inaccuracy in recall. I doubt this time frame.

Response: The SGPALS is designed to capture a person´s general PA level in the last year. Please see our reference nr 10: Saltin & Grimby, 1968, Circulation. The lack of knowledge on the validity of this question used in adolescents is part of the justification for our study.

Reviewer comment 6: For discussion, four brief paragraphs are not enough. Findings of the current study should be further explained with support from previous studies. In addition, findings of the current study should be compared with previous studies for consistency and inconsistency.

Response: Thank you. We have expanded our discussion. Please see line 290-344.

Reviewer comment 7: When talking about the “Strengths and limitations” of the current study, it is better that future studies should be mentioned. For example, the last point at Lines 268-270.

Response: We have added a section called “implications for public health and future studies”, see line 396-400. 

Reviewer comment 8: The last section of the study is not written in a well-summarized and clear way. Needs to be further revised to make it clear.

Response: This section has been rewritten (line 388-391).

 

Reviewer 2:

Reviewer comment 1: With validity and reliability, we are generally building evidence that supports the use of a measure for a set of people, in a given situations, with reference to specific outcomes from the measure. My preference is that we talk about “building evidence for the validity of XX” and not “we are validating XX”. I understand that this is more of an individual preference, but single studies with specific groups and only 1 outcome are often used to support using measures in other groups with a different outcome from that measure. Part of this is because we often talk in our results and discussions about “validating XX”, which implies the instrument is good, not that it has support for use in similar contexts. 

Response: Thank you. We have accordingly changed the phrasing in the paper.

Reviewer comment 2: For the models in table 3 and table 4 (correlation and regression) we need some indication of the distributions of the 4 SGPAL categories in those groups. Similar to how you show us girls vs boys. You could do this with an expansion of table 2. Just add 4 columns to right with each SGPAL category. The reason we want to know this is to understand the differences better. If the Overweight BMI category has 90% of people in categories 1 and 2 then the differences, we see in Table 4 (8.97 vs 5.33) per category need to be interpreted in this context. 

Response: Thank you. We have added the suggested columns to table 2.

Reviewer comment 3: You might need to look at all the scatter plots and visually inspect for outliers driving correlation up or down. If you did this, mention it in the methods. 

Response: Thank you. We performed such scatter plots to confirm not having extreme outliers. This is now mentioned in the methods section; see line 216-217.

Reviewer comment 4: Not needed but you might consider adding the relationship between SGPAL and a “meeting recommendation” outcome computed with Accelerometer MVPA or steps. You could show % in each SGPAL group getting more or less than 60 minutes of MVPA per day. Just more support for using SGPAL in the context of public health recommendations. 

Response: Thank you. We have included this in Table 3 (previous submission Table 5), line 259-260.

Reviewer comment 5: Also not need but you could add to results (and evidence) by looking at accelerometer and SGPLAY score differences by group (sex, BMI, edu…). To see if SGPLAY identifies the same difference as accelerometer outcomes. o Example: Do we see the same difference in MVPA and SGPLAY score between girls and boys? � Significance and percent difference � Maybe it would look like this • Boys have 43 minutes of MVPA and average SGPLAY score of 3.1 • Girls have 37 minutes of MVPA and average SGPLAY score of 2.8 • Boys have 16% higher MVPA minutes and 11% higher SGplay scores both are statistically significant. •Supplement table 4 with figures to show group differences. This would be 15 plots in a supplement so we can see the change with each SGPAL level by group. Like figures I added below, but you would have each group (girls/boys) plotted, with different figures for each grouping (sex, BMI, education) and PA outcome (steps, CPM, MVPA) 

Response: We choose to not include such plots at this time, as this paper is part of a PhD thesis which in its present form meet the objectives and rationale for the thesis. Although a good idea, this requires treating the SGPALS as a continuous variable using the mean for each strata, which we believe may introduce a false accuracy level.

Reviewer comment 6: You did some very nice work with the evidence and results for different groups (Sex, BMI, Health), but the discussion of this (line 229 to 233) is lacking. There is literature in adults and children about how some demographic factors may impact validity and reliability evidence for a given questionnaire. Dig in a little more in this area and really contribute to this area of research. 

Response: Thank you! We have performed a new literature search to find evidence on questionnaire responses by different demographic variables. We have now included this in our introduction to provide a more thorough rationale for investigating our secondary aim. Please see line 92-106. We have also included a paragraph of this in our discussion. Please see line 325-344.

Specific Comments: 

Line number / Priority / Comment

Reviewer comment 7: 39-40 moderate Correlations of 0.40 and 0.35 are fairly close for data like this. Consider removing “moderately” and “weakly” to describe magnitude. This will apply through the manuscript (discussion also).

Response: Thank you. We have removed this accordingly. 

Reviewer comment 8: 70-74 Low I would prefer talking about the strength of the validity evidence rather than “is thoroughly validated”. As an example… your aim is to build validity evidence for using the SGPALS in Norwegian adolescents. 

Response: Thank you. We have accordingly changed our phrasing. 

Reviewer comment 9: 94 High Did excluded participants (n=427) differ from included (n=572)? Demographics and SGPALS ratings. You mention this in the discussion (line 241-247), but this information should be in the methods. 

Response: Thank you. We included this under methods. Please see line 131-134.

Reviewer comment 10: 116 Low May need another few words or sentence with more info about “study program”. What does this mean for high schoolers in Norway? 

Response: Thank you. We have provided more information in school programs in Norway. Please see line 158-162.

Reviewer comment 11: 150 High Either in methods or start of results/table 2 it would be good to report average wear times (hours per day and Days per week). You should also confirm that none of the comparison groups (Sex, BMI, others in table 3) have differences in wear hours per day. Wear time at the day level is likely correlated with minutes of MVPA and Steps at ~0.25-0.40 level. If wear time is different this would impact the values, you are using to assess validity evidence in this group.

Response: Thank you. We have performed analysis on wear time by different demographic variables. This is now included in supplementary Table 1. Additionally, please see line 208-212 for statistical outputs. 

Reviewer comment 12: Table2 or 175 Moderate Need to have % participants at each SGPAL level. Numbers are in table 5, but should likely be presented at start of results. Same for Mean (SD) for all PA outcomes (MVPA, CPM, Steps) 

Response: Thank you. To address this, and particularly your comment nr 17 (see below), we have changed Table 5 (ANOVA) to Table 3, and removed Table 4 (linear regressions). Table 3 (correlation analyses) is now labelled Table 4.

Reviewer comment 13: Table 2/175 Moderate Because a lot of your evidence is built around difference for various demographic variables, it would be beneficial to comment on how much those groups overlap. For example, Do BMI categories and Parent education overlap. Participants in lower education group tend to have higher BMI values.

Response: Thank you. There were no differences in distribution of BMI categories between parents´ education groups (p=0.20) and between sexes (p=0.22). Those being overweight and obese were more likely to report lower self-reported health status (p<0.001), while there were no differences in self-reported health between parental education groups (p=0.28). This can be added to the manuscript if desired. 

Reviewer comment 14: 178 Low I realize you have a range for describing the strength of the correlations, but 0.35 and 0.40 are very close in this context. Jumping from “weak” to “moderate” is over statement.

Response: Thank you. We have removed our phrasing regarding magnitude in the correlations. 

Reviewer comment 15: Table3 Low Rather than “**”for almost all correlations with indicators for p-value, it would be better for your story to only highlight the differences that are important (what you focus on in text). For example 

BMI CPM Steps MVPA

Under 0.43 0.35 0.38

Over 0.27 0.32* 0.20*

You could pick two indicators. One for group difference for same PA outcome (bold) and one for difference across PA outcome within a group (*)

Response: Thank you. This is accordingly changed per your recommendation. Please see Table 4 (previous submission Table 3).

Reviewer comment 16: 187/table 4 Moderate Is this information from a regression model, different than the ANOVA model for results reported in table 5? May need to explain this model in stats section.

Response: Thank you. Please see our response to reviewer comment 18.

Reviewer comment 17: Table 4 Moderate What about intercepts, this tells us if the rate of change differed, but not the average starting value. This would be important for group differences

Response: Thank you. Please see our response to reviewer comment 18.

Reviewer comment 18: Table 4 High My guess is they are not all linear. I think you need to provide the 15 plots in a supplement so we can see the change with each SGPAL level by group. Like figure below, but you would have each group (girls/boys) plotted, with different figures for each grouping (sex, BMI, education) and PA outcome (steps, CPM, MVPA)

Response: Thank you. As you show with the included figures below, these outputs are not linear, and it may be misleading to present unstandardized beta coefficients from linear regressions. Therefore, we have removed this table from this study, while keeping the mean differences by SGPALS groups (ANOVA). 

Reviewer comment 19: 188/189 Moderate This model may only be linear for steps (see plots below). For MVPA there is definitely a bigger jump moving from grp3 to grp4

Table5 Low I like this, but I think three box plots (one for each PA outcome) would have more impact for the reader when emphasizing the differences in activity across the 4 SGPAL ratings.

Response: Thank you. Please see our response to reviewer comment 18. Additionally, we have created Box plots of CPM, steps and MVPA for the four different groups. Please see line 252.

Reviewer comment 20: 207 Low Similar to rest of literature comparing SR to Objective measures

Response: Thank you – we agree. 

Reviewer comment 21: 215 Low Not sure you mean bias

Response: Thank you. We have removed the sentence about PA volume and intensity, as we do not have specific data on these measures. Thereby, the sentence on biases is deleted.

Reviewer comment 22: 216 While it may not be a very accurate estimate of volume, table 5 suggests that you might get pretty good estimates of volume, especially if you also accounted for sex, BMI, edu, and rating of health. You could check the precision of these estimates by looking at the measured accelerometer outcomes versus those predicted from the regression model. This would give you average errors and a way to quantify bias and average prediction error. For example, MVPA from SGPALS rating vs MVPa from accelerometer, MVPA from SGPALS rating and Demographics vs MVPA from accelerometer

Response: Thank you. Please see our response to reviewer comment 18.

Reviewer comment 23: 221/222 This point is important, but would be more impactful is you gave us some of the values from the adult data (26,28) for comparison. Example… If in adults the difference in steps across the 4 groups is similar (1000 steps) this would be nice to see. If adults show a similar pattern but magnitude is different (15 min of MVPA vs 8) this would also be interesting and important.

Response: Thank you. We have provided comparison between our study and adult studies. Please see line 321-323.

Reviewer comment 24: 232 Why? Do we see this in what you present in tables

Response: Thank you. We have removed this statement.

Reviewer comment 25: 249 - 256 I don’t think you need this. No one will need you to justify using acceleometers as gold standard. You also used three estimates from accelerometer which quantify different but overlapping parts of the PA puzzle. This is strength! 

Response: Thank you, we removed this paragraph.

Reviewer comment 26: 263 Would be better here, or in methods, for you to present the ICC for your data. ICC for all three outcomes over all days measured.

Response: We chose to use Spearman’s rho which is a measure of rank correlation. ICC typically assesses agreement between data in terms of absolute validity or reliability. We appreciate the advantage of ICC to assess agreement between for example multiple days; however, we do not currently have access to such data (please see comment 27). 

Reviewer comment 27: 265/266 Did you look at the levels over days (1 vs 2,3,4,5,6,7) ? You could look at this an see if you think there was reactivity.

Response: Unfortunately, due to technical limitations we do not currently have access to day-by-day data. 

Reviewer comment 28: 274-275 I think you need to provide the error levels and bias in this prediction at the individual and group levels before saying that volume and intensity should not be estimated.

Response: Thank you. In Table 3, we have now included mean and 95%CI, and also provided box plots for assessing error level at the individual level. We agree that we do not provide results to substantiate any conclusion on volume and intensity and have changed the conclusion accordingly.

---

## [Decision Letter · Decision Letter 1]

13 Jan 2022

PONE-D-21-01038R1Criterion validity of the Saltin-Grimby Physical Activity Level Scale in adolescents. The Fit Futures Study.PLOS ONE

Dear Dr. Beldo,

Thank you for submitting your manuscript to PLOS ONE. After careful consideration, we feel that it has merit but does not fully meet PLOS ONE’s publication criteria as it currently stands. Therefore, we invite you to submit a revised version of the manuscript that addresses the points raised during the review process.

We look forward to receiving your revised manuscript.

Kind regards,

Ru Zhang

Academic Editor

PLOS ONE

Journal Requirements:

Reviewers' comments:

Reviewer's Responses to Questions

**Comments to the Author**

1. If the authors have adequately addressed your comments raised in a previous round of review and you feel that this manuscript is now acceptable for publication, you may indicate that here to bypass the “Comments to the Author” section, enter your conflict of interest statement in the “Confidential to Editor” section, and submit your "Accept" recommendation.

Reviewer #3: All comments have been addressed

2. Is the manuscript technically sound, and do the data support the conclusions?

Reviewer #3: Yes

3. Has the statistical analysis been performed appropriately and rigorously? 

Reviewer #3: Yes

4. Have the authors made all data underlying the findings in their manuscript fully available?

Reviewer #3: No

5. Is the manuscript presented in an intelligible fashion and written in standard English?

Reviewer #3: Yes

6. Review Comments to the Author

Reviewer #3: It appears the authors have done a good job of responding to the previous reviewers comments, and in general have presented a useful data for the validity of the scale in question in adolescent samples. As a result I only feel the need to provide a few basic comments:

1. The content of the discussion could be expanded. Specifically, in the discussion the authors note the generally weak correlations between the objective measures and the SGPALS. Given this is a vital element of a validation study I think it requires more discussion. Potentially the authors could cite some other validation papers for similar PA measures and note that the current findings are not that different to previous validation papers findings on studies like the IPAQ. I think this is an important point to make that the mediocre, ie, the weak correlations between self-report and objective PA measures found here are not unique. This is especially important given this measure seems to be much shorter than its alternatives but has produced similar findings, and could be viewed as a strength of the measure.

2. I also think more elaboration on future research and usefulness is needed. The implications given are true, but more could be done here. For example comparing the SGPALS to other self-report measures, or comparing a similar validity study in adults to further probe if age effects the validity of the measure.

7. PLOS authors have the option to publish the peer review history of their article (what does this mean?). If published, this will include your full peer review and any attached files.

Reviewer #3: No

---

## [Author Response · Author response to Decision Letter 1]

8 Jul 2022

Ru Zhang

Academic Editor

PLOS ONE

We hereby submit the revised version of our manuscript entitled “Criterion validity of the Saltin-Grimby Physical Activity Level Scale in adolescents. The Fit Futures Study.” We wish to thank the reviewer for relevant and insightful comments and suggestions. A point-by-point response to the reviewers has been submitted together with the revised manuscript.

We apologize for the delay and look forward to continue the process with our study together with PLOS ONE.

Yours sincerely,

Sigurd Beldo

---

## [Decision Letter · Decision Letter 2]

10 Aug 2022

Criterion validity of the Saltin-Grimby Physical Activity Level Scale in adolescents. The Fit Futures Study.

PONE-D-21-01038R2

Dear Dr. Beldo,

We're pleased to inform you that your manuscript has been judged scientifically suitable for publication and will be formally accepted for publication once some minor errors needs to be revised (Please see the attached comment file from the Reviewer 4). 

Within one week, you'll receive an e-mail detailing the required amendments. When these have been addressed, you'll receive a formal acceptance letter and your manuscript will be scheduled for publication.

If your institution or institutions have a press office, please notify them about your upcoming paper to help maximize its impact. If they'll be preparing press materials, please inform our press team as soon as possible -- no later than 48 hours after receiving the formal acceptance. Your manuscript will remain under strict press embargo until 2 pm Eastern Time on the date of publication. For more information, please contact onepress@plos.org.

Kind regards,

Ru Zhang

Guest Editor

PLOS ONE

---

## [Editor Report · Acceptance letter]

23 Aug 2022

PONE-D-21-01038R2 

Criterion validity of the Saltin-Grimby Physical Activity Level Scale in adolescents.
The Fit Futures Study 

Dear Dr. Beldo:

I'm pleased to inform you that your manuscript has been deemed suitable for publication in PLOS ONE. Congratulations! Your manuscript is now with our production department. 

Kind regards, 

on behalf of

Dr. Ru Zhang 

Guest Editor

PLOS ONE